# PPAR-α Insufficiency Enhances Doxorubicin-Induced Nephropathy in PPAR-α Knockout Mice and a Murine Podocyte Cell Line

**DOI:** 10.3390/cells13171446

**Published:** 2024-08-28

**Authors:** Kohei Matsuta, Kazuko Kamiyama, Toru Imamoto, Izumi Takeda, Shinya Masunaga, Mamiko Kobayashi, Naoki Takahashi, Kenji Kasuno, Masanori Hara, Masayuki Iwano, Tadashi Toyama, Hideki Kimura

**Affiliations:** 1Department of Clinical Laboratory, University of Fukui Hospital, Fukui 910-1193, Japan; kmatsuta@u-fukui.ac.jp (K.M.); kazuko3191220@yahoo.co.jp (K.K.); timamoto@u-fukui.ac.jp (T.I.); adoi@u-fukui.ac.jp (I.T.); smasu@u-fukui.ac.jp (S.M.); ttoyama@u-fukui.ac.jp (T.T.); 2Division of Nephrology, Department of General Medicine, School of Medicine, Faculty of Medical Sciences, University of Fukui, Fukui 910-1193, Japan; imty@u-fukui.ac.jp (M.K.); ntakahas@u-fukui.ac.jp (N.T.); kasuno@u-fukui.ac.jp (K.K.); miwano@u-fukui.ac.jp (M.I.); 3Iwamuro Health Promotion Center, Niigata 953-0104, Japan; mhara@iwamurohp.jp

**Keywords:** PPAR-α, podocyte, glomerulosclerosis, autophagy, p62, AMPK, doxorubicin, apoptosis, caspase

## Abstract

Peroxisome proliferator-activated receptor-alpha (PPAR-α) and its exogenous activators (fibrates) promote autophagy. However, whether the deleterious effects of PPAR-α deficiency on doxorubicin (DOX)-induced podocytopathy are associated with reduced autophagy remains to be clarified. We investigated the mechanisms of PPAR-α in DOX-induced podocytopathy and tubular injury in PPAR-α knockout (PAKO) mice and in a murine podocyte cell line. DOX-treated PAKO mice showed higher serum levels of triglycerides and non-esterified fatty acids and more severe podocytopathy than DOX-treated wild-type mice, as evidenced by higher urinary levels of proteins and podocalyxin at 3 days to 2 weeks and higher blood urea nitrogen and serum creatinine levels at 4 weeks. Additionally, there was an increased accumulation of p62, a negative autophagy marker, in the glomerular and tubular regions in DOX-treated PAKO mice at Day 9. Moreover, DOX-treated PAKO mice showed more severe glomerulosclerosis and tubular damage and lower podocalyxin expression in the kidneys than DOX-treated control mice at 4 weeks. Furthermore, DOX treatment increased p-p53, an apoptosis marker, and cleaved the caspase-3 levels and induced apoptosis, which was ameliorated by fenofibrate, a PPAR-α activator. Fenofibrate further enhanced AMPK activation and autophagy under fed and fasting conditions. Conclusively, PPAR-α deficiency enhances DOX-induced podocytopathy, glomerulosclerosis, and tubular injury, possibly by reducing autophagic activity in mouse kidneys.

## 1. Introduction

Peroxisome proliferator-activated receptors (PPARs) are members of the nuclear hormone receptor superfamily of ligand-dependent transcription factors [1]. Notably, this superfamily includes three different subtypes (PPAR-α, PPAR-β/δ, and PPAR-γ), with distinct expression patterns, different ligand-binding specificity, and biological functions. Additionally, these PPARs form heterodimers with RXRα, bind to the peroxisome proliferator response elements (PPREs) located in the promoter region of target genes, and exert specific biological actions. PPAR-α and PPAR-β/δ are implicated primarily in lipid metabolism and in the regulation of cell proliferation and differentiation, whereas PPAR-γ plays a pivotal role in the regulation of adipogenesis and insulin sensitivity [1]. Importantly, the transcriptional role of PPARs is selectively promoted by various ligands according to their binding features. PPAR-α is the molecular target of endogenous long-chain fatty acids and exogenous hypolipidemic fibrates such as fenofibrate and pemafibrate [2,3]. Generally, PPAR-α regulates a set of genes associated with fatty acid β-oxidation and plays an essential role in the metabolic control of energy homeostasis [4]. Therefore, PPAR-α is predominantly expressed in tissues with high catabolising rates of fatty acids [5,6], such as liver, heart, and kidneys, especially the proximal tubules. In the kidneys, PPAR-α is present and moderately functions in podocytes in the glomeruli [7,8,9]. PPAR-α activation exerts anti-inflammatory and antifibrotic effects by interfering with inflammatory transcription pathways and promoting fatty acid oxidation [10,11,12]. For example, PPAR-α gene deficiency exacerbates functional and histological damage in diabetic mice [11], high fat-loaded mice [13], and aged mice [14]. Additionally, fibrate-induced activation of PPAR-α has been shown to mitigate renal injury in cisplatin-treated tubular cells [15], diabetic mice [12], and high fat-loaded mice [16]. Notably, recent findings suggest that PPAR-α enhances autophagy in mouse liver [17,18]. Moreover, fenofibrate-induced activation of PPAR-α enhances autophagy in murine renal tubular cells during starvation [15] and kidneys of fat-loaded mice [19].

Doxorubicin (DOX), an antitumour anthracycline, is widely used to treat several solid malignancies and lymphomas [20]. However, its clinical use is limited because of its toxic effects. For example, DOX can cause self-preserving glomerulopathy [21]. Additionally, DOX induces podocyte injury and increases glomerular permeability, leading to proteinuria and glomerulosclerosis [22,23,24]. Notably, the cytotoxic mechanisms of DOX include several derangements, such as DNA alterations, the production of reactive oxygen species, mitochondrial dysfunction, and dysregulation of autophagic function [20,24]. Rodent models of DOX-induced nephropathy have been generated to elucidate the mechanisms of podocytopathy, nephrosis, and glomerulosclerosis and evaluate the protective effects of target molecules on nephropathy [24,25]. However, the susceptibility to DOX-induced nephropathy varies considerably among mice strains due to genetic factors [26,27]. BALB and S129 mice are sensitive to DOX, as evidence by podocyte injury and glomerulosclerosis, whereas C57BL6/J mice are resistant [24,25]. Considering that podocytes have a high level of basal autophagy, which contributes to the maintenance of podocyte integrity [28], the suppression of autophagy can aggravate their damage. Recent studies have indicated that autophagy inhibition exacerbates DOX-induced podocytopathy [29,30].

PPAR-α deficiency may worsen DOX-induced nephropathy, especially podocyte injury and the resultant glomerulosclerosis, via reduced autophagic activity. Moreover, a previous study showed that DOX enhanced apoptotic damage in podocytes and exacerbated albuminuria in PPAR-α-deficient mice [31]. However, whether the deleterious effects of PPAR-α deficiency on DOX-induced podocytopathy are associated with reduced autophagy remains to be clarified. Therefore, this study aimed to investigate the mechanisms of PPAR-α in DOX-induced podocyte and tubular injury in PPAR-α knockout (PAKO) mice and a murine podocyte cell line (mPod cells). Specifically, we compared podocyte health, tubular injury, and autophagy between DOX-treated wild-type and PPAR-α-null mice and investigated whether PPAR-α activation can enhance autophagy and protect against DOX-induced podocyte damage.

## 2. Materials and Methods

### 2.1. Materials

Fenofibrate (F6020) and doxorubicin (D1515) were purchased from Sigma-Aldrich (St. Louis, MO, USA). Rabbit monoclonal antibodies against mouse cleaved caspase-8 (#8592), phosphorylated mouse p53 (p-p53; Ser15, #12571), phosphorylated AMPK (p-AMPK, #2535), and rabbit polyclonal antibodies against human cleaved caspase-3 (#9661) and LC3 (#2775) were purchased from Cell Signalling Technology (Danvers, MA, USA). A rabbit polyclonal antibody against human p62 (PM045) was purchased from Medical and Biological Laboratories Co., Ltd. (Nagano, Japan). Goat polyclonal synaptopodin antibody (sc-21537) was purchased from Santa Cruz Biotechnology (Paso Robles, CA, USA). Mouse monoclonal anti-human PCX antibody was purchased from Denka Seiken (Niigata, Japan), whereas rabbit polyclonal antibody against mouse β-actin (ab8227) was purchased from Abcam Inc. (Cambridge, UK). Horseradish peroxidase (HRP)-conjugated anti-rabbit, anti-mouse, and anti-goat immunoglobulin antibodies were purchased from Dako (Glostrup, Denmark). 

### 2.2. Cell Culture

An immortalised murine podocyte cell line (mPod cells), kindly provided by Dr Peter Mundel (Goldfinch Bio Inc., Cambridge, MA, USA), was cultured as previously described [32]. Briefly, mPod cells were cultured in RPMI 1640 medium containing 10% foetal calf serum (Sigma-Aldrich, St. Louis, MO, USA) in collagen type I-coated dishes in the presence of 10 U/mL of mouse recombinant interferon-γ (IFN-γ, Pepro Tech, London, UK) at 33 °C in a 5% CO_2_ atmosphere. To initiate differentiation and maturation, mPod cells were maintained in a growth medium without IFN-γ in 12-well plates coated with collagen type I at 37 °C for at least 2 weeks. Podocyte injury was induced by culturing matured mPod cells with medium containing 0, 1.0, and 5.0 μg/mL of DOX for 15 and 24 h. Thereafter, mPod cells were cultured with medium containing DOX (0 or 1.0 μg/mL) in the presence or absence of fenofibrate (10 or 50 μM) for 24 h to investigate whether PPAR-α activation can reduce DOX-induced podocyte injury. To investigate the effect of fenofibrate on the basal autophagy status, cells were treated with medium containing fenofibrate (0 or 50 μM) for 3 or 6 h. For starvation experiments, mPod cells were incubated for 3 or 6 h in fasting medium—that is, HBSS medium with Ca_2_^+^ and Mg_2_^+^ (Fujifilm Wako Pure Chemical Corp., Osaka, Japan) in the presence or absence of fenofibrate (50 μM), as previously reported [17]. All experiments were performed in triplicate, unless otherwise stated.

An immortalised mouse proximal cell line (mProx) was generated as previously described [33] and confirmed to exhibit a proximal tubular phenotype [33]. Cells were grown in modified K-1 medium (50:50 Ham’s F-12/DMEM; Wako Pure Chemical Industries, Ltd.) containing 10% FBS (Thermo Fisher Scientific Inc., Waltham, MA, USA) at 37 °C in a humidified atmosphere (5% CO_2_ and 95% air). mProx cells (passages 15–20) were seeded in 12-well plates, and the modified K-1 medium was renewed every 2 days until confluence.

### 2.3. Annexin V-FITC/Propidium Iodide (PI) Assay

To detect early apoptosis, we examined phosphatidylserine translocation from the inner to the outer plasma membrane using an Annexin V-FITC kit (MBL Co., Ltd., Nagoya, Japan) according to the manufacturer’s protocol. Briefly, mPod cells grown in 12-well plates were treated with or without DOX (1.0 µg/mL) for 24 h in the presence or absence of fenofibrate (50 μM). Cells were harvested by trypsinisation, washed with phosphate buffer, and incubated in the dark with a binding buffer containing annexin V and PI (a late-phase marker of apoptosis) for 15 min at room temperature. Finally, fluorescence-activated cells were sorted using a flow cytometer (BD FACSCanto II, BD Biosciences, San Jose, CA, USA) and FlowJo software version 7.6 (BD Biosciences). Notably, cell sorting was based on 10,000 events/sample to obtain adequate data.

### 2.4. TaqMan Real-Time PCR Assay

TaqMan real-time PCR was performed as previously described [34]. Unlabelled specific primers and TaqMan MGB probes (6-FAM dye-labelled) against mouse PPAR-α (Mm00440939_m1), LC3 (Mm00458724_m1), ATG12 (Mm00503201_m1), and p62 (Mm00448091_m1) were purchased from Thermo Fisher Scientific, Inc. Notably, the mRNA levels of each gene were normalised to that of β2-microglobulin, unless otherwise stated. The average mRNA level of each gene in the untreated cells was set at 1.0 unless otherwise stated.

### 2.5. Immunoblot Analysis

Whole lysates of mPod cells and renal tissues were prepared using RIPA buffer containing phosphatase inhibitors (Santa Cruz Biotechnology). Protein lysates (10 µg) were immunoblotted/transferred into membranes as previously described [34]. After blocking, the membranes were incubated with anti-p-p53 (1:1000), anti-cleaved caspases-3 (1:1000), anti-cleaved caspase-8 (1:1000), anti-p62 (1:1000), anti-LC3 (1:500), anti-p-AMPK (1:1000), ant-synaptopodin (1:5000), and anti-β-actin (1:6000) antibodies for 20 min at room temperature. Thereafter, the membranes were incubated with the appropriate horseradish peroxidase-conjugated anti-rabbit or goat immunoglobulin (1:1000) at room temperature for 1 h. Finally, proteins were detected using ECL reagents, and the levels of each protein were normalised to that of β-actin. The average amount of the target protein in untreated cells was expressed as an arbitrary value unless otherwise stated. Each target protein and the control marker, β-actin, were electrophoresed on different gels, transferred onto different membranes, and separately hybridised with the corresponding antibodies. Adequately cropped images of the original blots are shown in the figures in this manuscript, and full-length images of the original blots are included in the Appendix A.

### 2.6. Animals and Experimental Design

PPAR-α knockout (PAKO) and wild-type (WT) S129SvJ mice were purchased from Jackson Laboratory (Bar Harbor, ME, USA) for animal experiments. Blood samples were collected from 10-week-old male mice (WT mice, *n* = 9; PAKO mice, *n* = 7) before and after a 48-h fast to compare fasting-induced metabolic changes (lipid profiles in blood) between PAKO and WT mice; after which, the mice were sacrificed to evaluate macroscopic changes in the livers. To investigate the effect of PPAR-α gene deficiency on DOX-induced renal injury, 10–12-week-old male mice were divided into four treatment groups: WT mice treated with saline, WT mice treated with DOX, PAKO mice treated with saline, and PAKO mice treated with DOX. Mice were injected with an equal volume of DOX or saline (10 mg/kg) via the tail vein once on day 1. Spontaneously voided urine within 24 h was collected from each animal a day before DOX injection (day 0); 1, 3, and 4 days post-injection (days 1, 3, and 4); and 1, 2, 3, and 4 weeks post-injection. Additionally, the body weights of the mice were measured at the indicated time points above. The protein and podocalyxin levels were measured in all urine samples, and the albumin levels were determined in the urine samples at day 0 and weeks 1 and 2.

In the early stage (days 0–9), mice were sacrificed on day 9 to collect blood samples for the lipid profile and renal function analyses and kidney samples for immunohistochemical analysis. In the late stage (weeks 2–4), mice were sacrificed at weeks 2 and 4 to collect blood samples for the lipid profile and renal function analyses and kidney samples for histological analyses.

### 2.7. Biochemical Analyses of Blood and Urine Samples

The blood levels of creatinine (Cr), blood urea nitrogen (BUN), total cholesterol (TC), triglyceride (TG), and non-esterified fatty acids (NEFA) and urinary levels of creatinine and protein were measured using an automatic biochemical analyser (TBA-2000FR; Canon Medical Systems Corp., Tochigi, Japan). The urinary albumin levels (μg/mL) were measured using an Albuwell ELISA kit (Exocell, Philadelphia, PA, USA). The urinary podocalyxin levels (ng/mL) were measured using sandwich ELISA, as previously reported [35]. The urinary protein, albumin, and podocalyxin levels were standardised to the urinary Cr levels (mg/dL) and expressed as g/gCr, g/gCr, and mg/gCr, respectively.

### 2.8. Renal Histological Analyses

Briefly, mouse kidneys were fixed in 4% paraformaldehyde, paraffin-embedded, cut into 3-µm-thick sections, and stained with haematoxylin and eosin (H&E) or Periodic acid–Schiff (PAS) stain. Glomerulosclerosis and tubular necrosis in PAS-stained kidney sections were blindly assessed under a light microscope using >20 glomeruli per slide and >10 fields per slide, respectively. Glomerulosclerosis was scored according to an earlier reported criteria [36] with a slight modification as follows: 0 = normal, 1 = 1–50%, 2 =  51–75%, and 3 =  76–100% sclerosis, and the glomerulosclerosis score (GSS) for each section was calculated as GSS =  [(1  ×  N1)  +  (2  ×  N2)  +  (3  ×  N3) ]/(N0  +  N1  +  N2  +  N3), where Nx is the number of glomeruli with each given grade for a given section. Tubular necrosis (brush border loss, necrosis, or detachment of the proximal tubular cells and cast formation) was scored as previously described [37,38]: 0 = normal, 1 = 1–10%, 2 = 11–25%, 3 = 26–45%, 4 = 46–75%, and 5 = 76–100%.

### 2.9. Immunohistochemistry

Kidney samples (2-μm sections) from mice treated with saline alone, DCA alone, cisplatin alone, or cisplatin + DCA were fixed in 4% paraformaldehyde, paraffin-embedded, and immune-stained. Rabbit polyclonal antibodies against p62 (1:1000) and podocalyxin (1:200) were used as primary antibodies to determine p62 and podocalyxin localisation. Positive staining was detected using the EnVision^TM^ FLEX kit/HRP (DAB; Agilent Technologies Japan, Ltd., Tokyo, Japan) according to the manufacturer’s instructions. Thereafter, the sections were counterstained with haematoxylin and mounted. The p62 staining intensity was observed in each slide using a light microscope under low-power examination. Notably, the area of apparently strong staining (hot spot) was detected, and the total number of hot spots in the glomerular region and tubular regions were separately calculated for each slide. The results were considered the autophagy status in the two regions for each mouse kidney.

### 2.10. Statistical Analyses

All individual data are presented as dot plots with the mean ± standard error of the mean (SEM). Statistical significance between two or more groups was determined using Student’s *t*-test or one-way analysis of variance (ANOVA). Statistical significance was set at *p* < 0.05. All statistical analyses were performed using SPSS (IBM SPSS Statistics 24.0), and the graphs were created using GraphPad Prism 10.2.2 (GraphPad Software Inc., Richmond, VA, USA).

## 3. Results

### 3.1. Fasted PAKO Mice Shows Fatty Liver, Higher Serum Levels of NEFA, and Lower Serum Levels of TC

Considering that fasting enhances fatty acid oxidation in the liver under normal conditions, we subjected WT and PAKO mice to fasting for 48 h. A representative gross photograph of the liver of WT and PAKO mice after 48-h fasting is shown in Figure 1A. Macroscopic examinations showed apparent fatty changes in the livers of PAKO mice, with a pale or white colour, whereas no detectable fatty changes were observed in the liver of WT mice (Figure 1A). Additionally, a significant increase in the serum levels of NEFA, a significant decrease in TC levels, and a decreasing trend in the glucose levels in PAKO mice were noted after 48-h fasting compared to those in the WT mice (Figure 1B–D). Moreover, the PPAR-α mRNA levels were significantly lower in the liver and kidneys of PAKO mice than in WT mice (Appendix A). Furthermore, the LC3 mRNA levels were significantly lower in PAKO mice than in WT mice under fed and fasting conditions, although the LC3 mRNA levels were approximately 1.6–2.0-fold higher in both mouse strains under fed and fasting conditions (Appendix A). Overall, these results indicate that the oxidation of free fatty acids (FFA) is suppressed in fasted PAKO mice, resulting in lipid accumulation in the liver and increased TC and glucose consumption as energy sources.

### 3.2. DOX-Treated PAKO Mice Present More Severe Proteinuria and Hyperlipidaemia Than Dox-Treated WT Mice in the Early Stage

DOX treatment caused a gradual increase in the urinary levels of protein and podocalyxin in both PAKO and WT mice until day 7 (Figure 2A,B). Notably, the urinary protein and podocalyxin levels were significantly higher in DOX-treated PAKO mice than in DOX-treated WT mice at days 3, 4, and 7 (Figure 2A,B). Additionally, the urinary levels of albumin, an injury marker of glomerular basement membranes, were significantly higher in DOX-treated PAKO mice than in treated WT mice at weeks 1 and 2 (Appendix A). Moreover, DOX-treated PAKO mice had significantly higher serum TC, TG, and NEFA levels and BUN levels than DOX-treated WT mice at day 9 (Figure 2C–F). However, there was no significant difference in the serum Cr levels between PAKO and WT mice (Figure 2G). Collectively, these results indicate that DOX-treated PAKO mice are more susceptible to podocyte injury and severe hyperlipidaemia than DOX-treated WT mice.

### 3.3. DOX-Treated PAKO Mice Present More Severe Renal Dysfunction Than DOX-Treated WT Mice in the Late Stage

DOX treatment increased the urinary levels of protein and podocalyxin in both WT and PAKO mice, peaking at 1 and 2 weeks post-injection, respectively (Figure 3A,B). Notably, the urinary protein levels were significantly higher in DOX-treated PAKO mice than in WT mice at 1, 2, 3, and 4 weeks post-injection (Figure 3A). Additionally, DOX-treated PAKO mice had significantly higher urinary levels of podocalyxin than did the DOX-treated WT mice at 1 and 2 weeks post-injection but lower levels at 3 and 4 weeks post-injection (Figure 3B). Moreover, PAKO mice showed significantly higher levels of BUN, Cr, and TC than DOX-treated WT mice at 4 weeks post-injection (Figure 3C–E). However, there were no significant differences in the serum TG and NEFA levels between DOX-treated PAKO and WT mice (Figure 3F,G). Overall, these results indicate that DOX-treated PAKO mice were more susceptible to severe proteinuria, podocyte injury, and renal dysfunction than WT mice throughout the treatment period of 4 weeks.

### 3.4. DOX-Treated PAKO Mice Present Lower Autophagic Activity and More Severe Glomerulosclerosis and Tubular Damage Than DOX-Treated WT Mice

DOX-treated PAKO mice displayed moderate and severe glomerulosclerosis and tubular damage at 2 and 4 weeks post-treatment, respectively, whereas DOX-treated WT mice showed mild to moderate glomerulosclerosis and tubular damage at 2 and 4 weeks post-treatment (Figure 4A–F). Furthermore, DOX-treated PAKO mice displayed more severe glomerulosclerosis and tubular damage than WT mice at 2 and 4 weeks post-treatment (Figure 4A–F). Compared to that in DOX-treated mice, more intense and frequent hot spots of p62 staining, an indicator of low autophagy activity, were observed in the glomerular and tubular regions of DOX-treated PAKO mice on day 9 (Figure 4G–I). Glomerular visceral, parietal, and tubular epithelial cells were mainly stained for p62 (Figure 4G). In contrast, p62 staining was hardly detected in the kidneys of untreated PAKO and WT mice, suggesting only weak p62 expression under normal conditions (Figure 4G). Similarly, immunoblotting using cell lysates from renal tissues revealed a significantly higher p62 expression in the kidneys of DOX-treated PAKO mice than in those of DOX-treated WT mice at day 9 (Figure 4J). The baseline p62 mRNA levels were approximately 10% lower in the kidneys of PAKO mice than in those of WT mice (Appendix A). In contrast, there was no significant difference in p62 mRNA expression in the kidneys between DOX-treated WT and PAKO mice at day 9 (Appendix A). DOX-treated PAKO mice showed a significantly lower intensity of podocalyxin staining than DOX-treated mice at 2 and 4 weeks post-treatment (Appendix A), supporting the changes in the urinary podocalyxin levels in DOX-treated PAKO mice (Figure 3B). Collectively, these results indicate that PPAR-α deficiency suppresses autophagy in the glomeruli and tubules in the early stage and exacerbate glomerulosclerosis and tubular damage in the late stage in DOX-treated PAKO mice.

### 3.5. Fenofibrate, a PPAR-α Activator, Ameliorates DOX-Induced Apoptosis in mPod Cells

As shown in Appendix A, the mouse podocyte cell line (mPod cells) used in this study displayed unique morphological features, with synaptopodin expression specific to primary podocytes. DOX treatment increased cleaved caspase-3 expression in mPod cells in a time- and dose-dependent manner, although the fold increase (over 100-fold) in cleaved caspase-3 expression at 24 h was similar at DOX doses of 1.0 and 5.0 μg/mL (Appendix A). Based on these results, a DOX concentration of 1.0 μg/mL was selected for subsequent experiments, unless otherwise stated. As for a PPAR-α activator, fenofibrate was used to investigate the effects of PPAR-α activation on DOX-induced injury of mPod cells, because the reagent is a representative activator of PPAR-α [2,3] and was previously reported to enhance autophagy in a murine renal tubular cell line [15]. Then, fenofibrate treatment (50 μM) significantly decreased the expression of p-p53, cleaved caspase-3, and cleaved caspase-8 in mPod cells stimulated with DOX (1.0 μg/mL) for 24 h by 67, 54, and 30%, respectively (Figure 5A–C). Apoptotic cells were quantified via flow cytometry using Annexin V and PI staining. Fenofibrate treatment (50 μM) significantly reduced DOX-induced apoptosis in mPod cells from 23.9% to 12.8% in Q4 (an early phase of apoptosis; Figure 5D,E).

### 3.6. Fenofibrate Enhances Autophagy, Possibly via AMPK Activation in mPod Cells under Fed and Fasting Conditions

Fasting for 3 and 6 h decreased p62 expression and increased LC3 II/I expression in mPod cells (Figure 6A–D), suggesting that fasting enhanced autophagy in the cells as a typical physiological response. Fenofibrate treatment decreased the p62 levels and increased the LC3 II/I levels in mPod cells under fasting conditions for 3 and 6 h (Figure 6A–D). Additionally, fenofibrate treatment increased LC3 II/I expression in mPod cells under the fed conditions for 3 h (Figure 6C) and 6 h (Figure 6D), implying that fenofibrate activated autophagy in mPod cells even after feeding. Moreover, fenofibrate treatment increased the phosphorylation of AMPK, an inducer of autophagy, in mPod cells under the fed and fasting conditions for 6 h (Figure 6E). Furthermore, fenofibrate increased the mRNA expression of LC3 and ATG12 in mPod cells 24 h under the fed conditions (Appendix A), which can promote the activation of fenofibrate-induced autophagy.

## 4. Discussion

In this study, DOX-treated PPAR-α knockout mice showed higher serum TG and NEFA levels and urinary protein and podocalyxin levels and lower autophagy activity (p62 accumulation) in renal tissues in the early stage (3–9 days post-treatment) than did DOX-treated WT mice. In the late stage (until approximately 4 weeks), DOX-treated PAKO mice showed more severe proteinuria, glomerulosclerosis, and tubular damage than did DOX-treated WT mice. The urinary podocalyxin levels is a sensitive marker of podocyte injury in the early phase and a precise indicator of podocyte loss in the late phase. Additionally, DOX injection increased the expression of p-p53 and cleaved caspase-3 and induced apoptosis, which was improved by fenofibrate, a PPAR-α activator. Importantly, fenofibrate treatment enhanced AMPK activation and autophagy in mPod cells during feeding and fasting. Conclusively, PPAR-α deficiency enhances DOX-induced podocytopathy and tubulopathy, probably by reducing FFA oxidation and autophagic activity in mouse kidneys.

Notably, the inhibition of PPAR-α activation exacerbated DOX-induced podocytopathy and glomerulopathy in mice kidney and murine podocytes, possibly by reducing autophagy and decreasing FFA oxidisation. A recent study showed that autophagy and lipophagy activation was weaker in the liver of PPAR-α-deficient mice at starvation than that in those of wild-type mice [17]. Additionally, PAKO mice show elevated serum levels of NEFA and fatty liver after fasting for several days due to neutral lipid accumulation via decreased FFA oxidisation [39,40]. In the present study, DOX-treated PAKO mice showed significantly higher serum levels of TG and NEFA; lower autophagy activity in glomerular visceral, parietal epithelial, and tubular cells; and mild glomerulosclerosis in the early phase (day 9). Lipid accumulation and reduced autophagy in the early stage may be closely associated with the subsequent development of podocytopathy and tubular injury. Moreover, podocytes are highly susceptible to intracellular accumulation of FFA and damaged proteins caused by lipid overload and dysfunction of the protein degradation system, autophagy, and the ubiquitin–proteasome system [28,41,42]. Notably, podocyte-specific inhibition of autophagy aggravates podocytopathy induced by exposure to noxious substances [30,41,43]. DOX-induced podocyte injury was more severe in PAKO mice than in WT mice as early as 3 days post treatment, as evidenced by a significant increase in the urinary podocalyxin levels. Overall, these results indicate that podocalyxin is a sensitive marker of podocyte injury and that PAKO mice are highly vulnerable DOX-induced podocytopathy.

Autophagy inhibition in tubular epithelial cells aggravates cellular damage such as apoptosis [15,44,45,46], whereas urinary protein overload upregulates autophagy in tubular cells, which is an adaptive response to protect the cells [47]. This may explain why the intense accumulation of p62 in the tubular regions at the early stage led to severe tubular damage at the late stage in PAKO mice. Reduced autophagic activity in podocytes and tubular cells was associated with severe renal damage in PAKO mice. Compared to previous findings [31], the PAKO mice used in the present study showed more severe uraemia and renal histological damage, although the mice used in the two studies were the same strain but different ages (10–12 vs. 8–10 weeks old). Additionally, DOX-induced nephropathy is more severe in older mice than in younger mice [25]. Moreover, the mice in the two studies may have differed to some degree in gene background, affecting susceptibility to DOX [26,27].

The primary molecular mechanism of PPAR-α-induced autophagy during fasting involves the upregulation of autophagy-related genes such as LC3, Agt7, and Agt12 [17]. Endogenous ligands of PPAR-α, such as long-chain fatty acids, are enriched during fasting, leading to fasting-enhanced transcriptional activity. Fenofibrate, an exogenous PPAR-α activator, enhances the expression of autophagy-related genes during feeding and fasting by increasing transcriptional activity and activating AMPK, an inducer of autophagy [17,19]. In the present study, LC3 expression was significantly higher in WT mice than in PAKO mice under the fed and fasting conditions (Appendix A). DOX-induced p62 accumulation in the renal tissues was weaker in WT mice than in PAKO mice. In mPod cells, fenofibrate treatment augmented the mRNA expression of LC3 and Agt12 at 24 h of feeding and activated AMPK and autophagy at 6 h of feeding and fasting. Additionally, fenofibrate treatment enhanced autophagy and attenuated apoptosis in DOX-treated mPod cells at 24 h. A previous study showed reduced autophagy activity (p62 accumulation) in the kidneys of fasted PAKO mice and its association with reduced ciliogenesis in tubular cells [48]. Based on these findings, it could be concluded that PPAR-α deficiency causes severe DOX-induced podocytopathy in PAKO mice and mPod cells, partly via the suppression of autophagy.

Finally, DOX-induced renal injury can also be enhanced by hypoperfusion due to DOX-induced cardiomyopathy. Heart weight (HW) to body weight (BW) ratios (HW/BW ratios, mg/gBW) at week 4 were compared among the four mice groups, as shown in Appendix A. DOX-treated PAKO and WT mice did not significantly differ in HW/BW ratios (6.01 vs. 5.36 mg/gBW, *p* = 0.1085). In contrast, marginal to significant increases in the HW/BW ratios were observed in DOX-treated WT and PAKO mice compared to DOX untreated WT and PAKO mice (Appendix A). Considering that DOX treatment produced slight cardiac hypertrophy in PAKO and WT mice, DOX-induced cardiomyopathy may have aggravated renal injury to some degree. However, the cardiac damage may have had a minor effect on renal injury in DOX-treated PAKO mice, because there was no significant difference in HW/BW ratios between DOX-treated PAKO and WT mice.

Despite the promising findings, our study had a few limitations. For example, it remains unclear whether fenofibrate attenuates DOX-induced nephropathy in PAKO mice. Whether the reduction in the liver production of LC3 was responsible for kidney injury/podocyte dysfunction after doxorubicin exposure also remains to be clarified. Additionally, immunofluorescence and electron microscopy are necessary to elucidate the autophagy status of podocytes in DOX-treated WT and PAKO mice. Moreover, the mechanisms through which autophagy inhibition influences the cell-protective effects of fenofibrate remain to be clarified. Furthermore, although there was a decrease in fatty acid β-oxidation in PAKO mice, as evidenced by the increased serum levels of TG and NEFA, we did not investigate lipid accumulation in the renal tissues using fat staining.

Conclusively, PPAR-α deficiency upregulates the serum levels of TG and NEFA, suppresses intrarenal autophagy, induces podocytopathy in the early stage, and aggravates proteinuria and functional and histological damage in the kidneys of DOX-treated mice. However, fenofibrate treatment attenuates DOX-induced apoptosis in mPod cells, possibly by enhancing autophagy. Overall, these results indicate that PPAR-α deficiency enhances DOX-induced nephropathy, probably by reducing FFA oxidation and autophagic activity in mouse kidneys.

## Figures and Tables

**Figure 1 cells-13-01446-f001:**
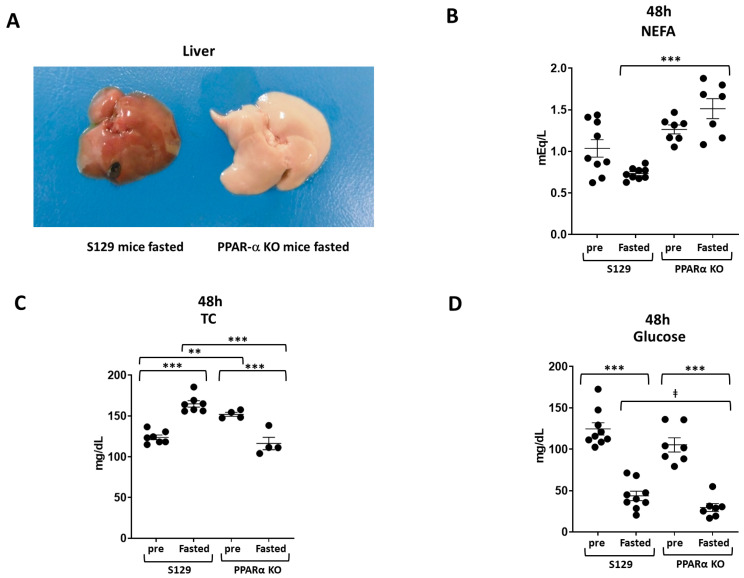
PPAR-α knockout (PAKO) mice showed fatty livers, higher serum levels of non-esterified fatty acids (NEFA), and lower serum levels of total cholesterol (TC) than wild-type (WT) mice after 48 h of fasting. Male PPAR-α-deficient S129 (PAKO) mice and WT S129 mice (10–12 weeks old) were fed rodent chow ad libitum as control mice and fasted for 48 h as fasted mice and then sacrificed for blood and liver sample collection. (**A**) Representative gross photograph of the liver of WT and PAKO mice after 48-h fasting. (**B**–**D**) Serum NEFA (**B**), TC (**C**), and glucose (**D**) levels were analysed. Individual data are presented as dot plots with the mean ± standard error of mean (SEM; *n* = 6–9). ** *p* < 0.01, and *** *p* < 0.001, according to one-way ANOVA with Tukey’s multiple comparisons test; ǂ *p* < 0.01, according to unpaired Student’s *t*-test.

**Figure 2 cells-13-01446-f002:**
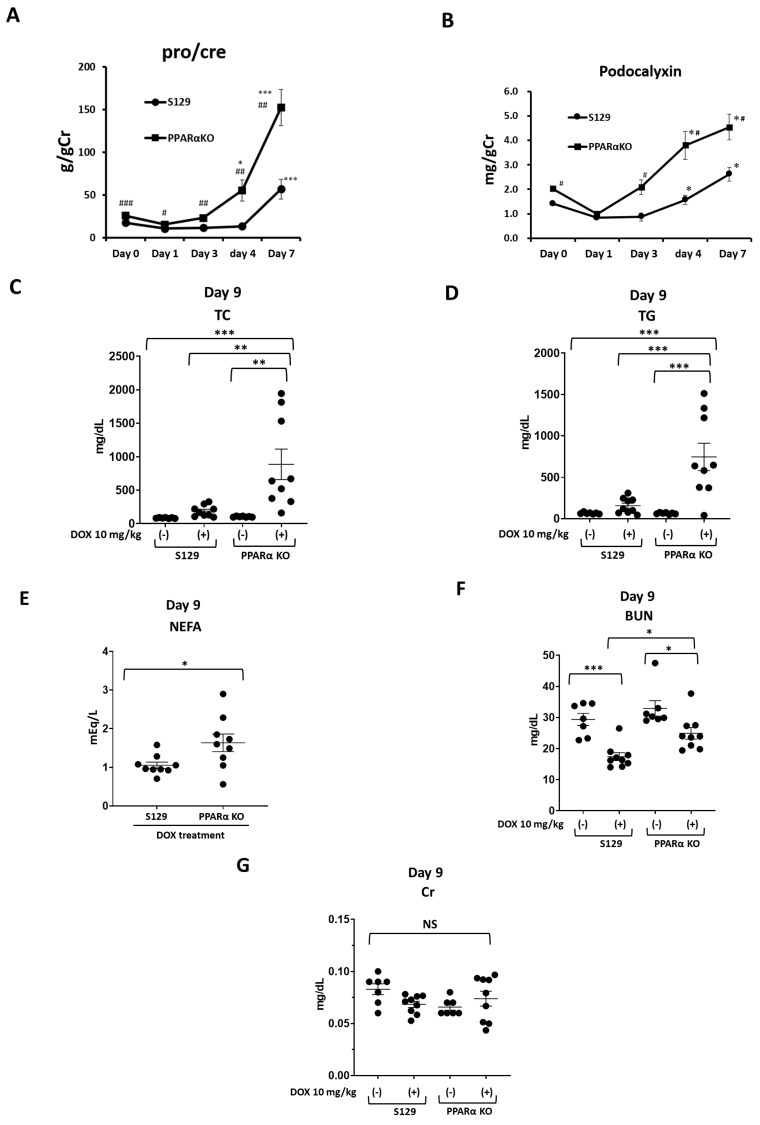
Doxorubicin (DOX)−treated PPAR-α knockout (PAKO) mice present more severe proteinuria and hyperlipidaemia than DOX-treated wild-type (WT) mice in the early stages. Male PPAR-α knockout S129 (PAKO) and WT S129 mice (10-12 weeks old, *n* = 7–9) were treated with DOX and sacrificed on day 9 to collect blood and kidney samples. Urine samples were collected on days 0, 1, 3, 4, and 7. The urinary protein (protein to creatinine ratio) (**A**) and podocalyxin (**B**) levels were measured at days 0, 1, 3, 4, and 7 as described in the Methods section. #, *p* < 0.05, ##, *p* < 0.01, and ###, *p* < 0.001: difference between the two groups at the same time point; * *p* < 0.05 and *** *p* < 0.001: differences from day 0 in each mouse group, based on an unpaired Student’s *t*-test. The serum levels of total cholesterol (TC) (**C**), triglyceride (TG) (**D**), non-esterified fatty acids (NEFA) (**E**), blood urea nitrogen (BUN) (**F**), and creatinine (Cr) (**G**) in WT and PAKO mice treated with and without DOX. Individual data are presented as dot plots with the mean ± SEM (*n* = 7–9). NS, not significant; * *p* < 0.05, ** *p* < 0.01, and *** *p* < 0.001 differences between the indicated groups based on one-way ANOVA with Tukey’s multiple comparison test (**C**,**D**,**F**,**G**) and an unpaired Student’s *t*-test (**E**).

**Figure 3 cells-13-01446-f003:**
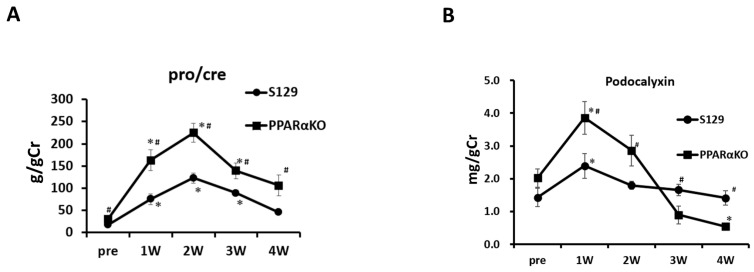
Doxorubicin (DOX)−treated PPAR-α knockout (PAKO) mice present more severe renal dysfunction than DOX-treated wild-type (WT) mice in the late stage. Male S129 (PAKO) and WT S129 (WT) mice (10−12 weeks old, *n* = 6–8) were treated with DOX and sacrificed at 4 weeks post-treatment to collect blood and kidney samples. Urine samples were collected on day 0 (pre) and at 1, 2, 3, and 4 weeks post-treatment. The urinary protein (protein to creatinine ratio) (**A**) and podocalyxin (**B**) levels were measured at day 0 and 1–4 weeks post-treatment, as described in the Methods section. # *p* < 0.05: difference between the two groups at the same time point; * *p* < 0.05: difference from day 0 (pre) in each mouse group, based on an unpaired Student’s *t*-test. The blood urea nitrogen (BUN) (**C**), creatinine (Cr) (**D**), serum total cholesterol (TC) (**E**), triglyceride (TG) (**F**), and non-esterified fatty acids (NEFA) levels (**G**) were measured in WT and PAKO mice with and without DOX treatment. Individual data are presented as dot plots with the mean ± SEM (*n* = 6–8). NS, not significant; * *p* < 0.05, ** *p* < 0.01, and *** *p* < 0.001 differences between the indicated groups based on one-way ANOVA with Tukey’s multiple comparison test.

**Figure 4 cells-13-01446-f004:**
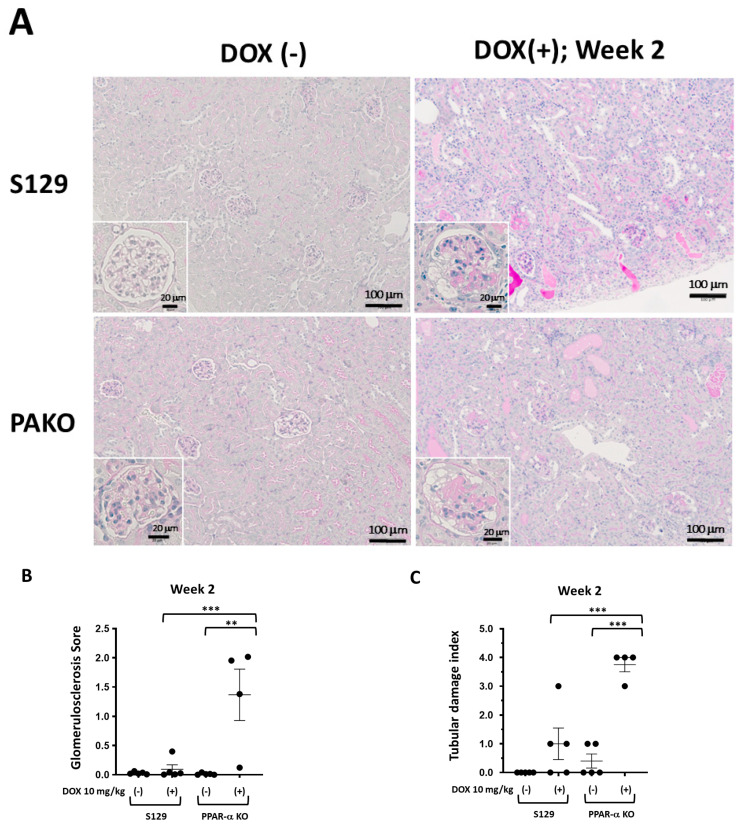
Doxorubicin (DOX)-treated PPAR-α knockout (PAKO) mice present lower autophagic activity and more severe glomerulosclerosis and tubular damage than DOX-treated wild-type (WT) mice. Male PPAR-α knockout S129 (PAKO) and WT S129 mice (10–12 weeks old) were treated with DOX and sacrificed to collect kidney samples at day 9 (*n* = 3–10) and weeks 2 (*n* = 4–6) and 4 (*n* = 4–16). Representative Periodic acid–Schiff (PAS)-stained kidneys of WT and PAKO mice with and without DOX treatment at 2 (**A**–**C**) and 4 weeks (**D**–**F**). Comparison of the glomerulosclerosis score (**B**,**E**) and tubular damage score (**C**,**F**) between WT and PAKO mice with and without DOX treatment. Representative immunohistochemical staining (**G**) of p62 in the kidneys of WT and PAKO mice with and without DOX treatment on day 9. p62-stained glomerular visceral and parietal epithelial cells and tubular epithelial cells are indicated by arrow heads, arrows, and asterisks, respectively. Inserts in the microphotographs show representative staining of glomeruli (scale bars, 20 μm). Comparison of the number of hot spots of p62 staining in the glomerular (**H**) and tubular (**I**) regions among the four groups. (**J**) Immunoblot analysis of p62 in the four groups on day 9. β-actin was used as a loading control. Representative blots are shown in the upper panels. Individual data are presented as dot plots with the mean ± SEM (*n* = 3–16). * *p* < 0.05, ** *p* < 0.01, and *** *p* < 0.001, indicating differences between the groups, based on one-way ANOVA with Tukey’s multiple comparisons test (**B**,**C**,**E**,**F**,**H**,**I**) and an unpaired Student’s *t*-test (**J**).

**Figure 5 cells-13-01446-f005:**
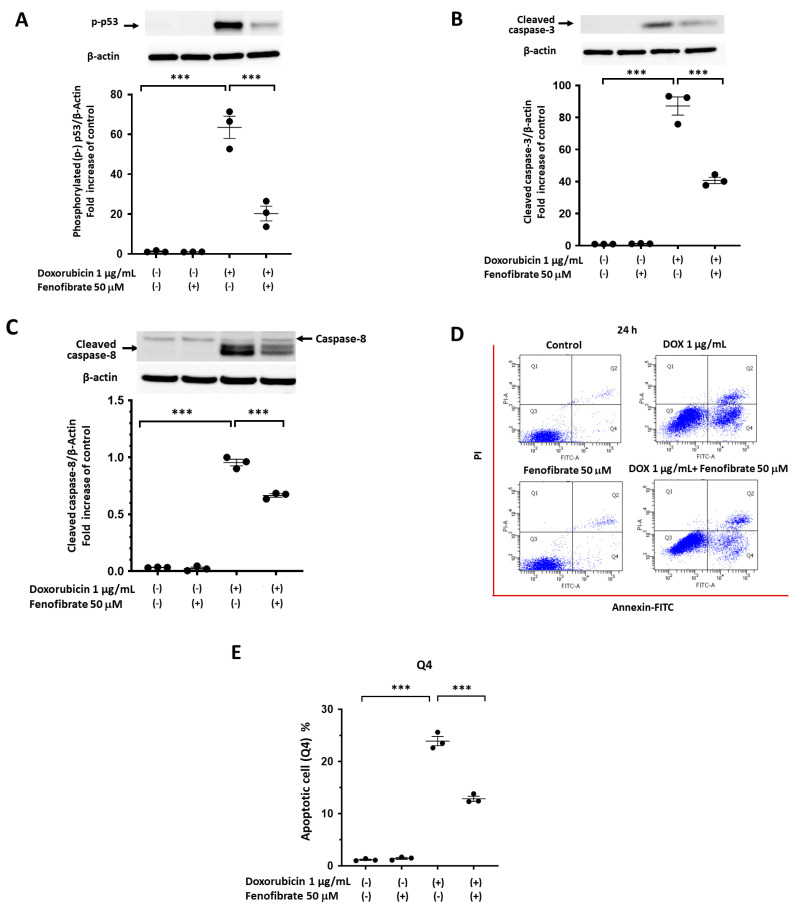
Fenofibrate, a PPAR-α activator, reduced doxorubicin (DOX)−induced apoptosis in mPod cells. mPod cells cultured on 12-well plates coated with collagen type I were treated with the growth medium (RPMI without INF-γ) containing DOX (0 or 1.0 µg/mL) in the presence or absence of fenofibrate (10 or 50 μM) at 37 °C for 24 h. The expression of phosphorylated p53 (**A**), cleaved caspase-3 (**B**), cleaved caspase-8 (**C**), and β-actin were determined by immunoblotting whole cell lysates. The amount of target protein was normalised to that of β-actin. (**D**,**E**) Harvested cells were stained with annexin-FITC and PI and analysed via flow cytometry using fluorescence-activated cell sorting, with the acquisition of a total of 10,000 events/sample. The percentages of Q2 and Q4 were calculated from the data for each well sample. Individual data are expressed as dot plots with the mean ± SEM of a representative experiment (*n* = 3). *** *p* < 0.001, different between the indicated groups, based on one-way ANOVA with Tukey’s multiple comparisons test.

**Figure 6 cells-13-01446-f006:**
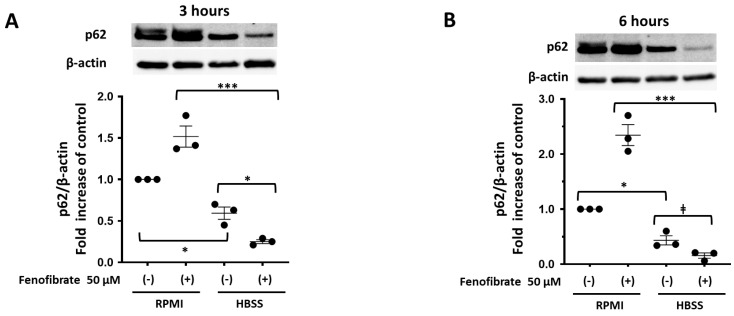
Fenofibrate enhances autophagy, probably via AMPK activation, in mPod cells under the fed and fasting conditions. (**A**−**E**) mPod cells were incubated for 3 or 6 h in the growth medium (RPMI without IFN-γ) or HBSS medium in the presence or absence of fenofibrate (50 μM). The expression of p62 (**A**,**B**), LC3 I and LC3 II (**C**,**D**), and phosphorylated (p-) AMPK (**E**) and β-actin were measured using immunoblot analyses and normalised to the levels of β-actin, except for LC3. Individual data are expressed as dot plots with the mean ± SEM of a representative experiment (*n* = 3). * *p* < 0.05, ** *p* < 0.01, and *** *p* < 0.001 indicate significant differences between the groups, based on one-way ANOVA with Tukey’s multiple comparisons test. ǂ *p* < 0.05 and ǂǂ *p* < 0.01 indicate significant differences between the groups, based on an unpaired Student’s *t*-test (**B**–**E**).

## Data Availability

The original contributions presented in the study are included in the article/Appendix A, and further inquiries can be directed to the corresponding author.

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
