# Peer review of "PPAR-α Insufficiency Enhances Doxorubicin-Induced Nephropathy in PPAR-α Knockout Mice and a Murine Podocyte Cell Line"

_cells, 2024, doi:10.3390/cells13171446_

Round 1

Reviewer 1 Report

Comments and Suggestions for Authors

Dear authors, thank you very much for allowing me to read your work. In your research, you present solid data based on knock-out mice for PPRA and murine podocyte cell culture, concerning the significance of autophagy in podocyte survival after exposure to doxorubicin. I have the following minor comments:

You performed experiments involving the liver and the effects of autophagy(Supplementary figure S2/3 - LC3 mRNA expression). Although you utilized male mice there is some minimal PPAR-α activity (liver<kidney). Fasting PPAR-α KO animals produce more LC3 mRNA in their kidney in comparison with the hepatic production of LC3 mRNA.  Was there any association concerning LC3 mRNA / protein production in the liver and renal dysfunction? In other words, is the reduction in the liver production of LC3 responsible for kidney injury/podocyte dysfunction after Doxorubicin exposure?

How can you control that the effects observed in the kidney of the mice you studied are not the effect of hypoperfusion due to doxorubicin-induced cardiomyopathy? Are the effects in cardiac tissue different among study groups (e.g. heart weight, extent of myocardial fibrosis)?

All the best.

Reviewer 2 Report

Comments and Suggestions for Authors

Authors show an interesting study, extensively analyzing the impact of PPAR gamma on DOX toxicity in animal model and podocyte cell line. Authors in a very detailed way explain how PPAR knockout affects DOX toxicity on a molecular and histological level. Additionally, fenofibrate effect was suggested, indicating its potential nephroprotective effect. 

Minor comments:

1) please explain in the abstract what are 'p62' and 'p-p53' or what is their significance;

2) why fenofibrate, not other fibrate derivatives was chosen in this study?

3) how 'averaged glomerulosclerosis score per glomeruli' was calculated? (Figure 4E)?

Comments on the Quality of English Language

Minor correcrions are needed (i.ex. damages L23; 'Sore' Y axis on Figure 4E).
